# FEATURE LEVEL INSTANCE ATTRIBUTION

## ABSTRACT

Instance attribution has emerged as one of the most crucial methodologies for model explainability because it identifies training data that significantly impacts model predictions, thereby optimizing model performance and enhancing transparency and trustworthiness. The applications of instance attribution include data cleaning, where it identifies and rectifies poor-quality data to improve model outcomes, and in specific domains such as detection of harmful speech, social network graph labeling, and medical image annotation, it provides precise insights into how data influences model decisions. Specifically, current instance attribution methods facilitate the identification of causal relationships between training data and model predictions. A higher Instance-level Training Data Influence value (IL value) indicates that the training data used for the computation play a more significant role in the model's prediction process. However, the current methods can only indicate that a training sample is important, but they do not explain why this sample is important. A feasible algorithm is urgently needed to provide an explanation for this behavior. This paper discovers that artificially manipulating the attribution score by modifying samples (e.g., changing a pixel value in image data) can significantly intervene in the importance of training samples and yield explainability results at the feature-level during the intervention process. The proposed Feature Level Instance Attribution (FLIA) algorithm assists in identifying crucial feature locations in training data that significantly impact causality. To avoid the frequent retraining of models for evaluation, we introduce an unlearning algorithm as an assessment method and provide detailed empirical evidence of our algorithm's efficacy. To facilitate future research, we have made the code available at: `https://anonymous.4open.science/r/FIIA-D60E/`.

## 1 INTRODUCTION

The development of artificial intelligence (AI) faces several challenges: improving model performance, defending against attacks, protecting data privacy, promoting fairness, enhancing interpretability, reducing computational requirements, and lowering annotation costs (Scherer, 2015; Hammoudeh & Lowd, 2024). Improving model performance is essential for accuracy and efficiency. Security measures are crucial to protect against attacks and data breaches. Addressing fairness prevents biases and social injustice. Enhancing interpretability builds trust and controllability. Reducing computational and annotation costs makes AI more accessible and practical. Failing to address these issues can significantly hinder AI development and application.

Training Data Influence Analysis (TDIA) evaluates the impact of individual training instances on a model's performance and predictions (Krishnan et al., 2016; Kong et al., 2021; Thimonier et al., 2022). By identifying influential data, this method can address key AI challenges. Removing problematic data improves performance and security by defending against poisoning and backdoor attacks (Shafahi et al., 2018; Oh et al., 2022; You et al., 2023). Influence analysis promotes fairness by detecting biases in data (Mehrabi et al., 2021). It also enhances interpretability by highlighting key training instances, making the model's decisions more transparent (Sui et al., 2021). Furthermore, it reduces computational requirements by selecting high-quality training subsets and lowers annotation costs by prioritizing significant unlabeled data, thereby improving efficiency and facilitating large-scale dataset creation (Braun et al., 2022).

Current TDIA algorithms, particularly those in the Gradient-Based Methods category like TracIn series, are relatively mature (Pruthi et al., 2020). However, these algorithms are limited to instance-

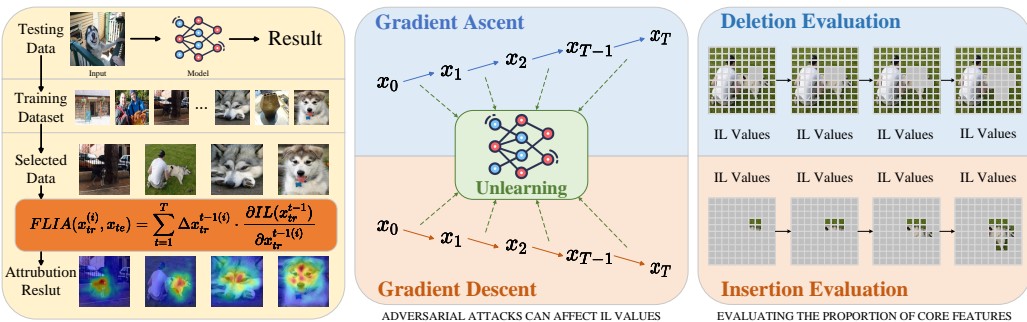

Figure 1: Flowchart of the FLIA process and Schematic Representation of Core arguments. The left section illustrates the FLIA workflow, while the middle section aligns with the arguments from Section 3 (Arguments 1 and 2) and the experimental designs of Section 4.2 (Experiment A) and Section 4.3 (Experiment B). To demonstrate that IL values can be altered, and that such alterations can affect the model's behavior, we employ unlearning techniques to assess this impact. The right section uses attribution results to evaluate the model, where gray occlusion areas represent adversarially attacked samples with occlusions applied to the original images.

level, meaning they can only assess the impact of an entire training sample on the model's decision. In other words, TDIA algorithms can only identify training samples that are highly correlated with the prediction but cannot explain why the sample has a high influence. Intuitively, if a TDIA algorithm cannot be explained or understood, we cannot trust that it has truly identified the most influential training samples. For example, as shown in Figure 2, TDIA algorithms may find backdoor attack samples but fail to determine which specific trigger caused the backdoor attack. In such cases, further analysis of these samples becomes difficult and requires substantial manual judgment to identify the problematic elements (such as the additional human costs to compare images to find triggers). If the trigger is not visually obvious Nguyen & Tran (2020), it is hard to distinguish between backdoor samples and supportive samples used in training. Based on this, our curiosity lies in identifying which features within a sample are key to its influence. Furthermore, we aim to develop a fine-grained influence assessment method capable of determining each feature's influence on the training data.

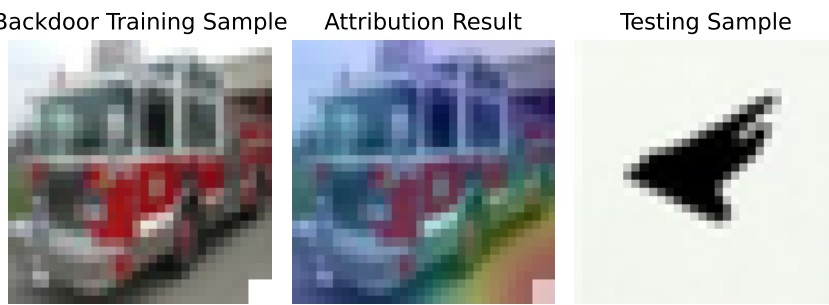

Figure 2: Backdoor Training Sample, Attribution Result, and Testing Sample

To achieve this, we devised a novel method to modify Instance-level TDIA values (referred to as IL values) by adding very small perturbations to the samples (Pruthi et al., 2020). To avoid repeatedly retraining the model to evaluate the impact of training data, we utilized unlearning algorithms to design a new evaluation method (Graves et al., 2021; Thudi et al., 2022; Liu et al., 2024). We found that these small perturbations could significantly influence the impact of training samples on model decisions (the larger the IL value, the greater the impact). Through the analysis of these perturbations, we rigorously derived the FLIA algorithm and provided strict proofs. The FLIA algorithm can capture all IL value changes and offer fine-grained feature-level TDIA. The flowchart of our FLIA method is shown in Figure 1, which illustrates how the algorithm computes influence changes, evaluates them through unlearning, and uses attribution results to assess model performance. Our contributions are as follows:

- We discovered that TDIA results could be altered by very small perturbations (for images, changes less than one pixel value), and these changes could significantly affect the model's decision-making process without altering the sample's confidence.

- We proposed the FLIA algorithm, a novel attribution algorithm that can obtain feature-level instance attribution results. To our knowledge, this is the first study to analyze the fine-grained impact of parts of the training data on model decisions.

- To facilitate further research and ensure experimental reproducibility, we provided a detailed derivation and rigorous mathematical proof of the FLIA algorithm's principles and have open-sourced all experimental code.

## 2 RELATED WORK

According to the study by Hammoudeh & Lowd (2024), TDIA methods can be categorized into retraining-based methods and gradient-based influence analysis methods, where gradient-based methods include both static and dynamic methods. In this section, we compare the principles, advantages, and disadvantages of different methods to clarify their application scenarios and limitations.

### 2.1 RETRAINING-BASED METHODS

Retraining-based methods assess the impact of each sample in the training data on the model output by removing each sample one by one and retraining the model. Leave-One-Out (LOO) (Weisberg & Cook, 1982) is the most classic retraining-based method. The core idea is to retrain the model after removing one training sample each time, and measure the impact of the sample by comparing the predictions of the new model and the original model on specific test instances. The main advantage of LOO is that it is intuitive and can accurately measure the influence of training instances. However, the computational cost of LOO is very high. For large datasets or complex models, the process of removing samples one by one and retraining the model is very time-consuming and resource-intensive, making it impractical for real-world applications.

### 2.2 GRADIENT-BASED METHODS

Gradient-based methods provide efficient influence analysis by analyzing the gradient impact of training data on model parameters and prediction results. Static methods and dynamic methods are the two main types.

#### 2.2.1 STATIC METHODS

Influence Functions (Koh & Liang, 2017) are a classic static method used to estimate the impact of small changes in training data on model parameters and prediction results. By slightly adjusting one instance in the training data and using gradient and Hessian matrix information, Influence Functions approximate the impact of this adjustment on model predictions. The main advantage of Influence Functions is that they do not require retraining the model. However, the Hessian matrix is computationally intractable for large model parameters and can only be approximated. Additionally, static methods are mainly applied to models at their final state, thus assuming the model is converged, which is not always the case in practice.

#### 2.2.2 DYNAMIC METHODS

TracIn (Pruthi et al., 2020) and HyDRA (Chen et al., 2021) are two main dynamic methods. TracIn tracks the gradient changes of each instance during the training process, records the gradient information at multiple time points, and accumulates these gradient changes to estimate the dynamic impact of each training instance on the prediction results of specific test instances. Its advantage is that it can dynamically capture the impact of training instances on the model, particularly suitable for deep learning models. HyDRA unfolds the test loss hypergradient concerning training data weights, comprehensively evaluating the contribution of training data to test data points. HyDRA simplifies the computation process by omitting the Hessian term, improving computational efficiency and performing well in handling noisy training data, but may introduce errors in certain cases.

## 2.3 OTHER RELEVANT METHODS

Fast Influence Functions (Guo et al., 2020) and LeafInfluence (Sharchilev et al., 2018) are two other important methods, both of which, along with mentioned Influence Functions (Koh & Liang, 2017), represent different versions of Influence Function-based approaches.. Fast Influence Functions achieve significant computational efficiency improvements through simple modifications to traditional Influence Functions. They mainly reduce computational complexity by narrowing the search space using k-nearest neighbors (kNN) algorithm and optimizing inverse Hessian-vector product estimation. Fast Influence Functions are suitable for large-scale datasets but may have some approximation errors. LeafInfluence is specifically designed for decision tree models, estimating the specific impact of each training instance on model predictions by analyzing the leaf nodes of decision trees. LeafInfluence is computationally efficient and applicable to single decision trees and ensemble models but is limited to decision tree models and not applicable to other types of models.

## 2.4 FEATURE ATTRIBUTION METHODS

Feature attribution methods aim to calculate the contribution of individual input features to model decisions, and they can be broadly divided into gradient-based and perturbation-based approaches. Gradient-based methods, such as Integrated Gradients (IG)(Sundararajan et al., 2017), compute attributions by integrating gradients from a baseline to the input, with extensions like Baseline Integrated Gradients (BIG)(Wang et al., 2021) and Adversarial Gradient Integration (AGI) introducing adversarial baselines and non-linear paths, respectively, to enhance robustness and accuracy. More advanced methods like More Faithful and Accelerated Boundary-based Attribution (MFABA)(Zhu et al., 2024) use second-order Taylor expansions to improve the efficiency of attributions, while AttEXplore(Zhu et al., 2023) focuses on incorporating model parameter information to refine feature importance. Despite the efficiency of gradient-based methods, they often suffer from sensitivity to model parameters and poor robustness to input perturbations, limiting their reliability in tasks such as insertion and deletion metrics. On the other hand, perturbation-based methods, like LIME (Ribeiro et al., 2016) and SHAP (Lundberg, 2017), work by modifying or removing input features and observing the effects on the model's output. LIME approximates local decision boundaries with surrogate models, while SHAP uses Shapley values from game theory to offer globally consistent attributions. Although these perturbation-based methods provide model-agnostic and interpretable explanations, they are computationally expensive and may lack robustness in complex data scenarios. Both approaches predominantly focus on local explanations for individual samples, making them less suitable for addressing the broader issue of TDIA.

Additionally, the TDIA introduced above is limited to the instance level, meaning it can only analyze the impact of a single sample on training (typically analyzing the association between the presence of one training sample and the model's decision on a single test sample at a time). The FLIA algorithm proposed in this paper can achieve feature-level analysis, i.e., analyze the contribution of each feature dimension within training samples to the TDIA.

## 3 METHOD

In this section, we will introduce the specific details of the FLIA algorithm. The core logic behind FLIA is to **observe the contribution of different dimensions of a sample during the process of modifying IL values**. To ensure that this core logic holds, we need sufficient experimental results to support three arguments.

- **Argument 1:** The IL values can be modified. This will be analyzed and proven in Section 4.2.

- **Argument 2:** We need to demonstrate that modifying IL values can directly affect the influence of a training example on a prediction made by the model. This will be analyzed and proven in Section 4.3.

- **Argument 3:** We must ensure that modifying IL values does not alter the inherent properties of the sample (such as the semantic information of the image or the model's confidence in the sample). This will be analyzed and proven in Section 4.4.

We will first introduce the instance-level algorithm TracIn used for TDIA and analyze the stability of the TracIn algorithm and the impact of minor sample modifications on TracIn results. Then, we will introduce how to use these perturbations to obtain feature-level instance attribution results.

## 3.1 INSTANCE-LEVEL TRAINING DATA INFLUENCE ANALYSIS

We first introduce the calculation process of IL values and the underlying derivation. From the derivation, we observe the connection between the IL values and the influence of a training example on a prediction made by the model.

Let $f$ represent the neural network, $w$ represent the parameters of the neural network, and $f(x;w)$ represent the output of the neural network for sample $x$ and parameters $w$. $L$ denotes the loss function, which typically represents the fit quality; the lower the loss function, the better the fit. For simplicity, we abbreviate the Instance-level TDIA algorithm as IL.

In Pruthi et al. (2020), the core principle of TracIn is **to observe the impact of training sample $x_{tr}$ on test sample $x_{te}$ after updating the parameters using $x_{tr}$.** The model's decision performance on test sample $x_{te}$ can be represented by the loss function $L(f(x_{te};w),y)$.

$$L(f(x_{te};w^t),y) - L(f(x_{te};w^{t-1}),y) \approx (\Delta w^{t-1})^\top \cdot \frac{\partial L(f(x_{te};w^{t-1}),y)}{\partial w^{t-1}} \tag{1}$$

As shown in Equation 1, by performing a first-order Taylor expansion on $L(f(x_{te};w^{t-1}),y)$ at time $t-1$, we can observe the impact of parameter changes on the test sample. The parameter change $\Delta w^{t-1}$ can be obtained by updating the parameters using the training sample $x_{tr}$. Under parameter $w^{t-1}$, the parameter update $\Delta w^{t-1}$ with gradient descent using training sample $x_{tr}$ is $\eta_t \cdot \frac{\partial L(f(x_{tr};w^{t-1}),y)}{\partial w^{t-1}}$. To observe the participation of training samples throughout the training process, we consider all parameter states during the training process, resulting in Equation 2.

$$
\begin{aligned}
IL(x_{tr}, x_{te}) &= \sum_{t=1}^{T} L(f(x_{te};w^t),y) - L(f(x_{te};w^{t-1}),y) \\
&= \sum_{t=1}^{T} \eta_t \cdot \underbrace{\frac{\partial L(f(x_{tr};w^{t-1}),y)}{\partial w^{t-1}}}_{g^t(x_{tr})}^\top \cdot \underbrace{\frac{\partial L(f(x_{te};w^{t-1}),y)}{\partial w^{t-1}}}_{g^t(x_{te})} \\
&= \sum_{t=1}^{T} \eta_t \cdot g^t(x_{tr})^\top \cdot g^t(x_{te})
\end{aligned}
\tag{2}
$$

Using Equation 2, we obtain the IL result. For convenience, we abbreviate the two gradients in Equation 2 as $g^t(x_{tr})$ and $g^t(x_{te})$. This result indicates the impact of the training sample $x_{tr}$ on the test sample $x_{te}$ over multiple training processes. Here, we summarize that the calculation of the IL value considers the impact on the loss function with and without the test sample $x_{te}$ at different stages of training. Since the loss function is the most direct way to reflect the model's prediction results, we can observe the influence of a training example on the model's prediction directly through the IL value.

## 3.2 PERTURBATION OF IL VALUES

The effectiveness of IL has been thoroughly validated in prior work Pruthi et al. (2020); Hammoudeh & Lowd (2024), so we do not delve into further details here. We observed that IL results can be easily perturbed by introducing minimal modifications to the samples (for image tasks, this typically involves altering each pixel by a very small amount). Notably, such perturbations do not alter the semantic information of the original images or affect the model's predictions. For instance, as demonstrated in Section 4.4, even when IL values experience significant shifts, the model's confidence in predicting the perturbed samples remains unchanged. This discussion aligns with **Argument 3**.

$$x_{tr}^t = x_{tr}^{t-1} \pm \eta \cdot \text{sign}\left(\frac{\partial IL(x_{tr}^t, x_{te})}{\partial x_{tr}^t}\right) \tag{3}$$

We update the training sample $x_{tr}$ using Equation 3, where $x_{tr}^0$ denotes the initial state of $x_{tr}$. The step size $\eta$ is set to $\frac{1}{2550}$ (as each pixel is normalized to a granularity of $\frac{1}{255}$), and the number of updates is set to 10. With this configuration, the maximum perturbation after 10 updates is constrained to within a single pixel value, which reason is that we found this small perturbation is already sufficient to induce significant changes in the IL value while maintaining the model's confidence. If we were to perturb more than a single pixel value, the IL value might change too drastically. When the perturbation direction is negative, the goal is to reduce the IL output, thereby decreasing the influence of the training sample on the test sample. We observe that over 10 iterations, the IL value decreases by at least 50%. Conversely, when the perturbation direction is positive, the goal is to enhance the training sample's influence on the test sample, with some cases showing an increase in IL by over 34 times after 10 iterations. This analysis supports **Argument 1**.

Additionally, we found that the pixel values of the samples and the model's confidence in those samples remained nearly unchanged during the perturbation process, as demonstrated in Section 4.2. Furthermore, we performed unlearning on the same sample under different IL values and observed the impact on the model's prediction before and after unlearning. This revealed a clear correlation between IL changes and the influence of the training sample on the model's decision-making, which we discuss in Sectionn 4.3. This finding suggests that adjusting the IL value can either enhance or weaken the influence of a training sample on the model's decision, supporting **Argument 2**.

### 3.3 FEATURE-LEVEL INSTANCE ATTRIBUTION

In this section, we introduce how the FLIA algorithm determines the importance of each feature dimension and provide the corresponding derivation process. To maintain simplicity in the derivation process, we abbreviate $IL(x_{tr}, x_{te})$ as $IL(x_{tr})$, and perform a Taylor expansion on $x_{tr}$ as the independent variable at time $t$:

$$IL(x_{tr}^t) = IL(x_{tr}^{t-1} + \Delta x_{tr}^{t-1}) = IL(x_{tr}^{t-1}) + \Delta x_{tr}^{t-1} \cdot \frac{\partial IL(x_{tr}^{t-1})}{\partial x_{tr}^{t-1}} + \mathcal{O} \tag{4}$$

where $\mathcal{O}$ represents higher-order infinitesimals, indicating that the approximation is accurate up to first-order terms, with higher-order terms contributing insignificantly for small perturbations. And $\Delta x_{tr}^t = \pm \eta \cdot \text{sign}\left(\frac{\partial IL(x_{tr}^t, x_{te})}{\partial x_{tr}^t}\right)$. Considering each moment:

$$\begin{cases} \sum_{t=1}^T IL(x_{tr}^t + \Delta x_{tr}^t) = \sum_{t=1}^T \left( IL(x_{tr}^t) + \Delta x_{tr}^{k\top} \cdot \frac{\partial IL(x_{tr}^{t-1})}{\partial x_{tr}^t} + \mathcal{O} \right) \\ x_{tr}^{t+1} = x_{tr}^t + \Delta x_{tr}^t \end{cases} \tag{5}$$

We finally derive the core formula for Feature-Level Instance Attribution:

$$FLIA(x_{tr}, x_{te}) = IL(x_{tr}^T) - IL(x_{tr}^0) = \sum_{t=1}^T \Delta x_{tr}^{t-1\top} \cdot \frac{\partial IL(x_{tr}^{t-1})}{\partial x_{tr}^{t-1}} \tag{6}$$

The contribution of the $i$-th dimension feature in $x_{tr}$ to the IL can be derived as:

$$FLIA(x_{tr}^{(i)}, x_{te}) = \sum_{t=1}^T \Delta x_{tr}^{t-1(i)} \cdot \frac{\partial IL(x_{tr}^{t-1})}{\partial x_{tr}^{t-1(i)}} \tag{7}$$

In the process of calculating $\frac{\partial IL(x_{tr}^{t-1})}{\partial x_{tr}^{t-1(i)}}$, only $g^t(x_{tr})^\top$ is related to the sample $x_{tr}$, while $g^t(x_{te})$ acts only as a weight. This means that $\frac{\partial IL(x_{tr}^{t-1})}{\partial x_{tr}^{t-1(i)}}$ evaluates the second-order curvature of the sample with respect to the parameter space manifold, establishing a connection between the parameters and

the sample. It is worth noting that the above derivation process proves that any change in $x_{tr}$ leading to a change in the IL value will inevitably be captured by the FLIA algorithm. Moreover, the sum of the importance of all feature dimensions equals the change in the IL value. This also implies that as long as the change in the IL value is meaningful, the attribution results will be able to distinguish the contribution of each feature to the IL value.

# 4 EXPERIMENTS

## 4.1 EXPERIMENTS SETTING

We conducted experiments on the following datasets: CIFAR-10 (Krizhevsky et al., 2009), CIFAR-100, GTSRB (The German Traffic Sign Recognition Benchmark) (Houben et al., 2013), and SVHN (The Street View House Numbers) (Netzer et al., 2011). To ensure reproducibility and reliability of the results, a fixed random seed of 0 was used in all experiments. The attack steps were set to 10, and the learning rate was set to 1/2550. We used two model architectures: ResNet-18 (He et al., 2016) and DenseNet-121 (Huang et al., 2017). We conducted all experiments via two NVIDIA A 100 graphics cards.

The specific settings are as follows:

- CIFAR-10: Randomly selected 100 images per class from the training set and 10 images per class from the test set, totaling 10,000 samples.
- CIFAR-100: Randomly selected 10 images per class from the training set and 10 images per class from the test set, totaling 10,000 samples.
- GTSRB: Randomly selected 15 images per class from the training set and 15 images per class from the test set, totaling 9,675 samples.
- SVHN: Randomly selected 100 images per class from the training set and 10 images per class from the test set, totaling 10,000 samples.

## 4.2 EXPERIMENT A: ADVERSARIAL ATTACKS CAN AFFECT IL VALUES

Table 1: Changes in IL values and confidence under adversarial attacks across different datasets and models. The Confidence Change column represents the variation in confidence values for the true class of the training samples, and the IL Change column represents the variation in IL values.

| Dataset | Sample Number | Gradient Direction | ResNet-18 | | DenseNet-121 | |
|---|---|---|---|---|---|---|
| | | | IL Change | Confidence Change | IL Change | Confidence Change |
| CIFAR-10 | 10000 | Gradient Descent | -0.5097 | 0.0004 | -0.6158 | 0.0007 |
| | | Gradient Ascent | 2.5005 | -0.0099 | 2.2480 | -0.0121 |
| CIFAR-100 | 10000 | Gradient Descent | -0.7403 | 0.0059 | -0.7434 | 0.0042 |
| | | Gradient Ascent | 42.0748 | -0.1403 | 110.6082 | -0.1794 |
| GTSRB | 9675 | Gradient Descent | -0.6107 | 0.0414 | -0.7008 | 0.1132 |
| | | Gradient Ascent | 10.2616 | -0.0548 | 6.8963 | -0.0470 |
| SVHN | 10000 | Gradient Descent | -0.6705 | 0.0033 | -0.7023 | 0.0079 |
| | | Gradient Ascent | 4.7778 | -0.0499 | 5.5495 | -0.0844 |

The results in Table 1 demonstrate that adversarial attacks have a pronounced effect on IL values, while the corresponding changes in class confidence are minimal. Across all datasets and models, the IL values exhibit significant shifts under both gradient ascent (increasing influence) and gradient descent (decreasing influence). For example, on the CIFAR-100 dataset using DenseNet-121, the IL value increased by over 110 during gradient ascent, but the confidence only decreased slightly by 0.1794. This pattern is consistent across other datasets such as GTSRB and SVHN, where IL changes are substantial, while confidence variations remain small.

These findings suggest that adversarial perturbations are particularly effective at altering the model's sensitivity to specific training samples without causing drastic changes in its confidence in the true

class. This highlights the vulnerability of IL values to adversarial manipulation, where the influence of training data on model predictions can be significantly amplified or reduced, even when the model's overall certainty about its predictions is largely unaffected. The disproportionately large changes in IL during gradient ascent, particularly on datasets like CIFAR-100, indicate that adversarial attacks can exploit the model's inherent sensitivity to specific data points, leading to significant shifts in influence even when starting from a relatively small baseline.

This analysis supports the **Argument 1** that adversarial attacks primarily target the influence of individual training samples on model decisions, rather than directly modifying the model's confidence in its predictions.

### 4.3 EXPERIMENT B: THE IMPACT OF IL VALUES ON TRAINING INFLUENCE

In Experiment B, we aim to verify **Argument 2** that the correlation between the influence of IL values and the effect of a training example on the model's predictions. To avoid retraining the model each time to evaluate the Training Data Influence, we adopted the Gradient ascent (GA) unlearning method (Graves et al., 2021; Thudi et al., 2022; Liu et al., 2024). Unlearning was performed using the SGD optimizer with a learning rate of 0.01, momentum set to 0, weight decay set to 0, and other parameters kept at their default values. Each unlearning step involved inputting one training sample and its corresponding label. During each unlearning iteration, a deep copy of the model was made, and the cross-entropy was calculated based on the output of the input training data followed by gradient ascent.

To evaluate relevance, we define a new evaluation metric called the Confidence Difference Correlation Index (CDCI). CDCI calculates the covariance by using the attack steps and the confidence difference at each step of the model. The confidence difference at each step is the absolute value of the change in the model's confidence for a test sample when unlearning a single training sample. The larger the difference, the greater the influence that training sample has on the model's training process. This metric is used to observe whether the influence of a training example on a model's prediction changes along with the IL value. If the CDCI is greater than 0, it indicates a clear positive correlation, and the larger the value above 0, the stronger the correlation **(typically, a value greater than 0.5 indicates a strong correlation)**. We conducted experiments on the CIFAR-10, CIFAR-100, GTSRB, and SVHN datasets and recorded the Confidence Difference Correlation Index (CDCI). The results are shown in Table 2.

As shown in Table 2, the results of unlearning under adversarial attacks exhibit a significant positive correlation with IL values. This indicates that as the number of attack steps increases, the absolute value of the impact caused by unlearning (i.e., the absolute difference in class confidence before and after unlearning) becomes larger, representing a stronger relationship between unlearning and the influence

Table 2: Confidence Difference Correlation Index (CDCI) across different datasets and models under adversarial attacks.

| Dataset | CIFAR-10 | CIFAR-100 | GTSRB | SVHN |
|---------|----------|-----------|-------|------|
| CDCI | 0.6440 | 0.7958 | 0.7573 | 0.660 |

of training data on model predictions. This suggests that the influence of unlearning intensifies with the progression of adversarial attacks. Negative values represent gradient descent, which reduces IL values and thus decreases the impact caused by unlearning.

To further illustrate this correlation, we plotted the output difference curves against the number of attack steps (see Figures 3a-3d). These figures show that as the number of attack steps increases, the output differences gradually increase, indicating a growing influence of unlearning on model predictions. This further validates the strong correlation between IL values and the influence of training examples.

In summary, as long as the change in IL values is meaningful, the attribution results are necessarily meaningful. This experimental result supports the effectiveness of our proposed method in evaluating the impact of training data.

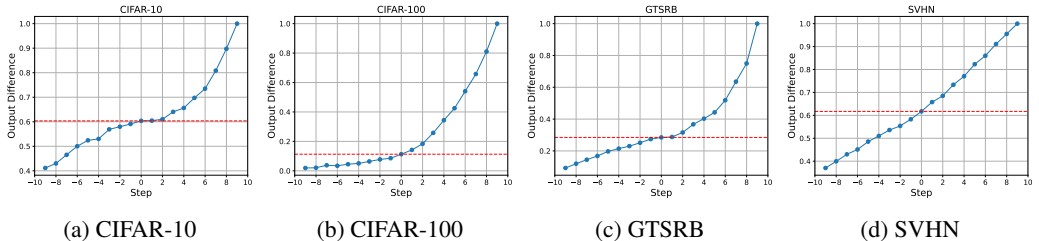

(a) CIFAR-10      (b) CIFAR-100      (c) GTSRB      (d) SVHN

Figure 3: Output difference vs. attack steps on ResNet-18. The red line represents the Output difference without attack.

Table 3: Insertion (INS) and Deletion (DEL) results across different datasets and models.

| Model | CIFAR-10 | | CIFAR-100 | | GTSRB | | SVHN | |
|---|---|---|---|---|---|---|---|---|
| | INS | DEL | INS | DEL | INS | DEL | INS | DEL |
| ResNet-18 | 0.6882 | 0.8807 | 0.4568 | 0.7773 | 0.8810 | 0.8835 | 0.9040 | 0.9794 |
| DenseNet-121 | 0.6024 | 0.8545 | 0.4582 | 0.7870 | 0.8922 | 0.9397 | 0.8868 | 0.9773 |

### 4.4 EXPERIMENT C: EVALUATING THE PROPORTION OF CORE FEATURES IN THE DATASET THROUGH IL VALUES

In Experiment C, we aim to evaluate the proportion of core features in the dataset through Insert-Deletion Analysis. Specifically, the insert operation gradually replaces the original image with the adversarial image, while the deletion operation gradually replaces the adversarial image with the original image. Each operation replaces 10% of the region according to the attribution results, performs a total of 10 replacements, and calculates the IL score to assess the impact of these changes on the model.

We conducted experiments on the CIFAR-10, CIFAR-100, GTSRB, and SVHN datasets using ResNet-18 and DenseNet-121 models. The specific experimental steps are as follows: First, we performed the insert operation, replacing 10% of the region of the original image with the adversarial result according to the attribution results each time, and calculating the IL score after each replacement, performing a total of 10 replacements. Second, we performed the deletion operation, replacing 10% of the region of the adversarial image with the original image according to the attribution results each time, and calculating the IL score after each replacement, performing a total of 10 replacements. To ensure data consistency and comparability, we normalized each replacement step by dividing by the maximum value and averaged the results over samples. The insert operation yielded the Insertion (INS) results, and the delete operation yielded the Deletion (DEL) results. If the INS result is less than the DEL result, it indicates that the attribution process is effective, and a larger gap indicates that the model utilizes fewer features from the training samples.

A smaller Insertion value indicates that a small number of attribution results can cover the entire IL attribution, suggesting that these core features occupy a smaller proportion during training but have a significant impact on model decisions. Conversely, a larger Insertion value indicates more concentrated key information. This can be used to assess whether the model fully utilizes the features in the training data and may also indicate the model's generalization ability. A model with a lower Insertion and higher Deletion value may have learned less concentrated key features during training, potentially leading to better generalization.

On the CIFAR-10 dataset, the ResNet-18 model's Insertion score (INS) is 0.6882, while the Deletion score (DEL) is 0.8807, indicating that the model can effectively reduce the IL value with fewer core feature changes. Similarly, the DenseNet-121 model on the same dataset has Insertion and Deletion scores of 0.6024 and 0.8545, respectively. For the CIFAR-100 dataset, the Insertion scores for the ResNet-18 and DenseNet-121 models are 0.4568 and 0.4582, while the Deletion scores are 0.7773 and 0.7870, respectively, further verifying the concentration of key information. The results on the GTSRB and SVHN datasets also show that the insert and delete operations can effectively identify and evaluate core features.

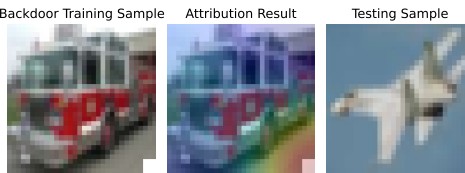

Figure 4: FLIA result on the normal training sample.

Figure 5: FLIA result on the backdoor attack training sample.

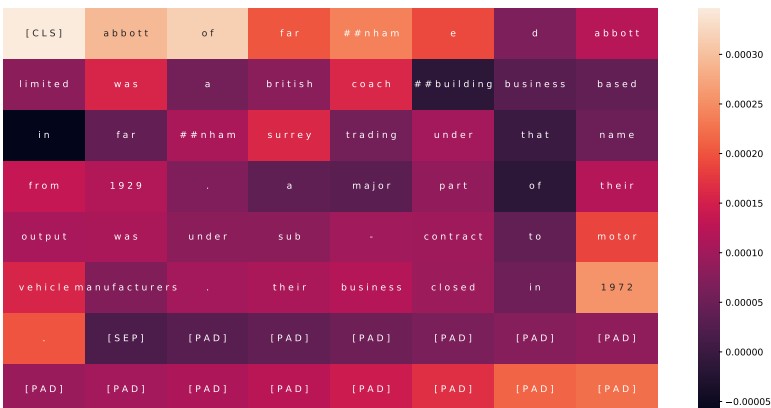

Figure 6: FLIA result on the NLP task.

Thus, using the FLIA method, we can see which regions of the training data are important for the test samples. Under normal circumstances, as shown in Figure 4, certain features in the training samples significantly affect the test samples. In the case of backdoor attacks, as shown in Figure 5, the trigger in the backdoor attack has the greatest impact on the test sample, indicating that our method could potentially detect backdoor attack samples. For any testing sample, the trigger in the backdoor attack plays a significant role in the model's final decision. Additionally, our method can also be applied in natural language processing, as shown in Figure 6, lighter colors indicate more important features. Key words such as "abbott," "british," and "1929" contribute most to the model's decisions, while padding tokens like "[PAD]" have minimal impact. These attribution results help identify the most influential information for the model's decision-making.

## 5 CONCLUSION

In this paper, we propose for the first time a feature-level method for Estimating Training Data Influence, named FLIA. This method identifies which specific features in the training samples have an impact on the model's decisions. We provide rigorous mathematical derivations and proofs to ensure its validity. Additionally, we designed three types of experiments to demonstrate the effectiveness and potential impact of the FLIA algorithm. A limitation of the current method is that, although theoretically effective, our validation relies on indirect evidence through the unlearning method. Further evaluation methods are needed to comprehensively verify the effectiveness of FLIA.

## CODE OF ETHICS AND ETHICS STATEMENT

All authors of this paper have read and adhered to the ICLR Code of Ethics[1] during the entire process of conducting this research and preparing the manuscript. We acknowledge and accept the ethical guidelines, which include promoting fairness, transparency, and integrity in AI research.

In our work, no human subjects were involved, and no new datasets were created that could raise privacy or security concerns. However, we acknowledge the potential risks associated with using model explainability and attribution techniques in sensitive applications such as healthcare or social networks. We have taken steps to ensure that the methods proposed do not exacerbate issues of bias or discrimination and are aligned with the broader goals of fairness and transparency in AI systems. There are no conflicts of interest or sponsorships to declare for this work.

## REPRODUCIBILITY STATEMENT

To ensure the reproducibility of our work, we have made the code, datasets, and detailed experimental setup available in the supplementary materials and at an anonymous link[2]. All models were trained using a fixed random seed, and the hyperparameters for each experiment are clearly described in Section 4 of the paper. We have also provided a comprehensive explanation of the theoretical results, including assumptions and proofs, in the appendix. Additionally, the steps taken to preprocess the datasets used in the experiments are included in the supplementary materials to facilitate replication of the experiments.

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

## A  Image Attribution Analysis

| Training Sample | Attribution Result | Testing Sample |
|---|---|---|
| 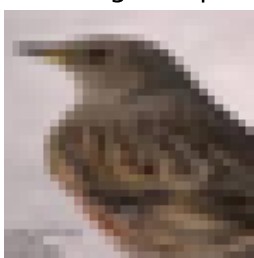 | 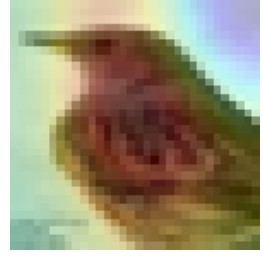 | 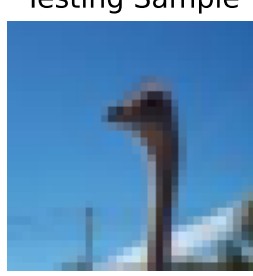 |

Figure 7: Normal Training Sample, Attribution Result, and Testing Sample

| Backdoor Training Sample | Attribution Result | Testing Sample |
|---|---|---|
| 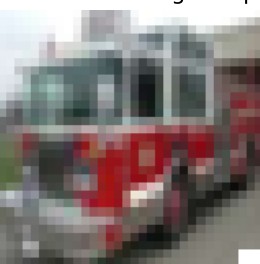 | 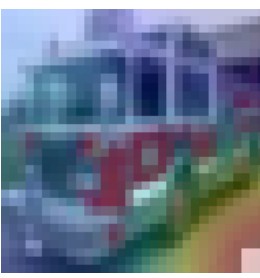 | 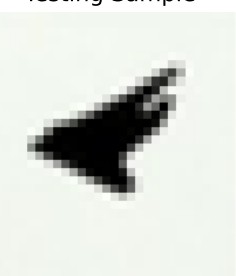 |

Figure 8: Backdoor Training Sample, Attribution Result, and Testing Sample

Figure 7 shows a comparison between a normal training sample, its attribution result, and the corresponding testing sample. The left side of the figure displays the training sample, which is an image of a bird. The attribution result in the middle highlights the areas of the image that the model focused on, with lighter colors indicating more important features. The right side of the figure shows the testing sample, which is another image of a bird. The attribution result helps to understand which features in the training sample are crucial for the model's decision on the testing sample.

Figure 8 presents a comparison between a backdoor training sample, its attribution result, and the corresponding testing sample. The left side of the figure shows the backdoor training sample, which is an image of a bus with a backdoor attack marker. The attribution result in the middle highlights the key areas the model focused on, with the white patch in the lower right corner being particularly significant. The right side of the figure displays the testing sample, which is an image of an airplane. The attribution result reveals which features in the backdoor training sample influenced the model's decision on the testing sample.

## B  NLP Attribution Analysis

Figure 9a shows the FLIA attribution result for a text about "schwan-stabilo." Lighter colors indicate more important features. The model attributes higher importance to key terms such as "schwan-stabilo," "german," "pens," "highlight," and "marker," which are crucial for understanding the context of the text.

Figure 9b presents the FLIA attribution result for a text about "goldilocks bakeshop." Key terms like "goldilocks," "philippines," "cakes," and "family business" are highlighted as important by the model, indicating their significant contribution to the text's meaning.

Figure 9c displays the FLIA attribution result for a text about "orange music electronic company." Important features include terms such as "orange," "amplifier," "british," and "distinctive sound," which the model considers crucial for the text's interpretation.

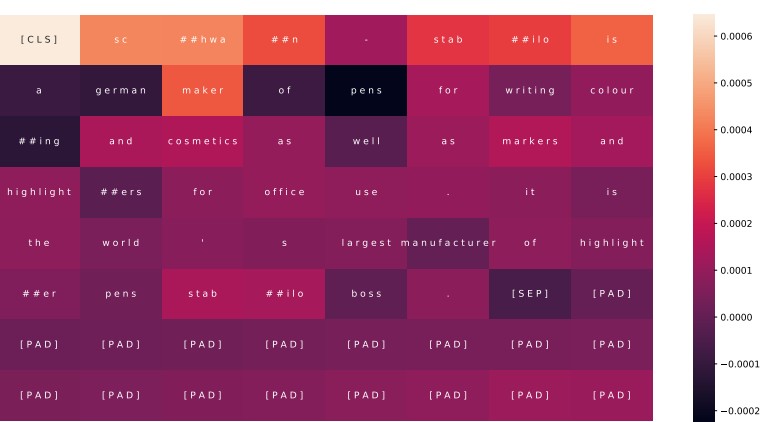

(a) FLIA Attribution Result for "schwan-stabilo"

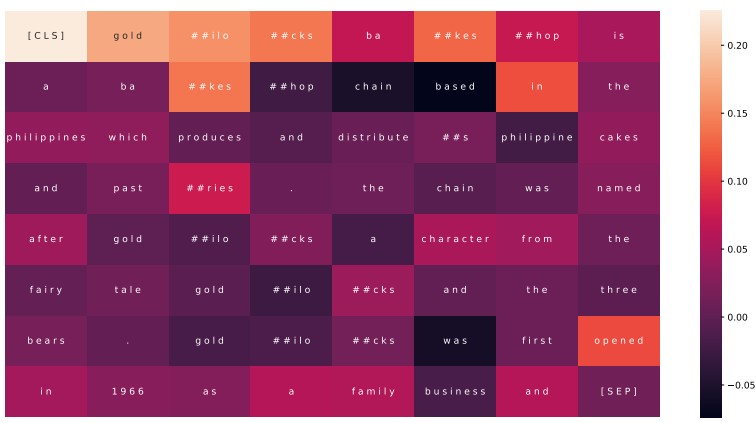

(b) FLIA Attribution Result for "goldilocks bakeshop"

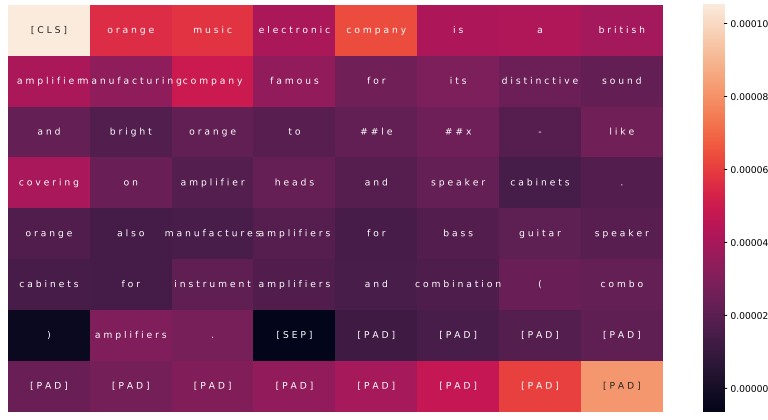

(c) FLIA Attribution Result for "orange music electronic company"

Figure 9: FLIA Attribution Results for Various NLP Tasks

