# OpenReview forum: "Feature Level Instance Attribution"
_ICLR.cc/2025/Conference — Submitted to ICLR 2025_

### Official Review · Reviewer_FniP · 2024-10-29

**Soundness:** 2
**Presentation:** 2
**Contribution:** 2
**Rating:** 3
**Confidence:** 4

**Summary:**

This paper discovers that artificially manipulating the attribution score by modifying samples can significantly intervene in the importance of training samples and produce explainability results at the feature-level during the intervention process. And the authors propose a new method called Feature Level Instance Attribution (FLIA) to locate crucial feature of training data that significantly impact causality. Although the theoretical analysis and experimental results are provided to support three arguments which ensure the core logic of FLIA, direct experimental validation of FLIA's effectiveness is lack. The practical applicability of FLIA requires further explanation and verification.

**Strengths:**

1) The paper discovers that TDIA results could be altered by very small perturbations, and these changes could significantly affect the model’s decision-making process without altering the sample’s confidence and proposes a new method called Feature Level Instance Attribution (FLIA).
2) The author provides rigorous mathematical derivation, introduces the calculation process of IL value, and derives the contribution of samples to IL value at the fine-grained feature level.

**Weaknesses:**

1) The validation of FLIA totally relies on indirect evidence，so the practical applicability of FLIA requires further explanation or experimental verification.
2) The experiment results are not convincing enough. In experiment A, the experimental results are not based on repeated experiments, and there was no significance analysis experiment. In experiment B, the evaluation metric used is self-defined and lack a clear formula definition. These makes the experimental results not convincing enough.
3) The writing of this paper needs to be strengthened. For example, Figures 1 and 2 are referenced in the wrong order. What’s more, the Figure 1 contains a lot of content but lacks detailed explanation, which makes it difficult to understand.

**Questions:**

In addition to the weaknesses I have already listed, I have the following questions:
1) The author claims that the FLIA implements feature-level analysis and is the first study to do so. However, this claim is questionable: in the related work section, existing feature attribution methods also aim to compute the contribution of features of training data to model decisions, so why do these methods fail to achieve fine-grained feature-level analysis? In order to support this argument more clearly, it is recommended that the authors provide a detailed comparison of FLIA with existing feature attribution methods, specifically clarifying the unique aspects of FLIA in analyzing the impact of features of these training data on model decisions, and the limitations of other methods in this context. Such a comparison will help readers better understand the innovation and contributions of FLIA.
2)  In the related work section, the authors introduce a categorization of TDIA methods, stating that they can be divided into retraining-based methods and gradient-based impact analysis methods, with the latter further consisting of both static and dynamic methods. However, upon reviewing this section, I found that the structure does not entirely align with the initial categorization presented. Specifically, the discussions in Sections 2.3 and 2.4 are not clearly linked to the stated categories. I recommend that the authors either revise their initial categorization to explicitly include the methods from Sections 2.3 and 2.4 or clarify how these additional methods relate to the TDIA framework they've outlined.

---

### Official Review · Reviewer_vywQ · 2024-11-02

**Soundness:** 1
**Presentation:** 2
**Contribution:** 3
**Rating:** 3
**Confidence:** 3

**Summary:**

The paper presents a novel approach to analyzing the training data influence of individual instances called the FLIA algorithm. Most notably it allows for understanding the feature values for each instance contributed towards the training data influence value which is not seen in previous works in the area. The paper presents three core arguments regarding how instance level testing data influence analysis values can me modified, methods for computing the IL values, and some experiments supporting the arguments.

**Strengths:**

The paper is well written with few writing issues, the introduction and related works provides a good high level overview of the task and why it is important. The need for novel feature level understanding of samples is well motivated.

**Weaknesses:**

While I believe the method has merit and is interesting, there are structural problems with the paper in its current state.

Major Issues:
- The concept of model confidence in a sample is introduced and used without any explanation but is used as a core piece of evidence supporting argument 1. The confidence difference correlation index is then introduced in the next section supporting argument 2 and its unclear whether these refer to the same metrics. Furthermore, the description for this metric doesn't provide a clear understanding of what it actually does, the sentence on lines 405 to 407 is particularly unclear
- The method section covers both prior works and the new proposed FLIA algorithm, it's not made explicitly clear what is original contribution and what are prior works. If the IL term as a product of two gradients is novel, please make this more explicit. In particular, the IL values is introduced in the context of previous works (98-103, 264-266) and the experiments consistently refer to IL values and its unclear when FLIA is actually being evaluated.
- I think the core arguments need to be more clearly defined and rigorous, what does it mean to be modified and why is it desirable at all? Additionally the terminology is inconsistent (i.e. Figure 1 uses the term 'altered' but argument 1 uses 'modified').

Minor issues:
- The code is difficult to understand, the code is quite dense without a single comment, even in the example notebook being provided.

Overall I think the paper needs to more clearly set out and define the aims, particularly the definitions and evaluation of the core arguments presented.

**Questions:**

- Were other random seeds tested for the experimental results?
- What is the new evaluation metric CDCI, how is it implemented, what are its properties?

---

### Official Review · Reviewer_jffN · 2024-11-04

**Soundness:** 3
**Presentation:** 2
**Contribution:** 2
**Rating:** 3
**Confidence:** 4

**Summary:**

This paper proposes to combine feature attribution and instance attribution and develops a feature-level instance attribution (FLIA) method. Specifically, the proposed method extends the TracIn method and identifies important features whose perturbation leads to large change in instance-level influence value. The proposed method is evaluated on a set of image classification settings with a variety of evaluation metrics.

**Strengths:**

1. This paper investigates an interesting problem that combines feature and instance attribution.

2. The proposed method is well-motivated.

3. The authors conducted extensive experiments.

**Weaknesses:**

1. This paper missed a directly relevant work [Pezeshkpour et al. 2022] that exactly combines feature and instance attribution, which significantly undermines the novelty of this work.

2. The clarity of the writing could be improved. For example, the arguments in Section 3 and their connection to the "core logic" aren't very clear to me.

3. The experiment setups, especially experiment A, should be better clarified. Section 4.2 directly talks about the experiment results without a clear description of the experiment setup, making it difficult to interpret the results.


References

[Pezeshkpour et al. 2022] Pouya Pezeshkpour, Sarthak Jain, Sameer Singh, Byron Wallace. Combining Feature and Instance Attribution to Detect Artifacts. Findings of ACL 2022.

**Questions:**

See Weaknesses.

---

### Meta-Review · Area_Chair_g2tY · 2024-12-21

**Metareview:**

The paper presents a method to combine feature attribution and instance attribution by perturbing samples to learn better explainability results. The paper demonstrates three experiment results on subselected datasets.

Strengths (based on reviewers' input):
 - Tackling an important problem with a well-motivated method
 - Paper includes three experiments and reproducibility code

Weaknesses
 - Concerns about the paper not engaging sufficiently with related work
 - Experimental setups were unclear
 - Experiment analysis does not include a significance calculation
 - In the methodology section, several concepts are not well explained, which makes the paper's arguments hard to follow.
 - Separation between contributions and prior work is at times unclear

Because of extensive concerns from the reviewers, I am advocating for a reject. I hope the authors are able to take the feedback and clarify their arguments for a stronger submission in future venues.

**Additional Comments On Reviewer Discussion:**

The authors did not include a rebuttal response.

---

### Decision · Program_Chairs · 2025-01-22

Reject